# Estimating Transcriptome Diversity and Specialization in *Capsicum annuum* L.

**DOI:** 10.3390/plants13070983

**Published:** 2024-03-29

**Authors:** Neftalí Ochoa-Alejo, M. Humberto Reyes-Valdés, Octavio Martínez

**Affiliations:** 1Departamento de Ingeniería Genética, Unidad Irapuato, Centro de Investigación y de Estudios Avanzados del Instituto Politécnico Nacional, Irapuato 36824, Guanajuato, Mexico; neftali.ochoa@cinvestav.mx; 2Department of Plant Breeding, Universidad Autónoma Agraria Antonio Narro, Saltillo 25315, Coahuila, Mexico; manuel.reyes@uaaan.edu.mx; 3Unidad de Genómica Avanzada (Langebio), Centro de Investigación y de Estudios Avanzados del Instituto Politécnico Nacional (Cinvestav), Irapuato 36824, Guanajuato, Mexico

**Keywords:** gene expression, RNA-Seq, Shannon information theory, gene specificity, transcriptome

## Abstract

Chili pepper fruits of the genus *Capsicum* represent excellent experimental models to study the growth, development, and ripening processes in a non-climacteric species at the physiological, biochemical, and molecular levels. Fruit growth, development, and ripening involve a complex, harmonious, and finely controlled regulation of gene expression. The purpose of this study was to estimate the changes in transcriptome diversity and specialization, as well as gene specificities during fruit development in this crop, and to illustrate the advantages of estimating these parameters. To achieve these aims, we programmed and made publicly available an R package. In this study, we applied these methods to a set of 179 RNA-Seq libraries from a factorial experiment that includes 12 different genotypes at various stages of fruit development. We found that the diversity of the transcriptome decreases linearly from the flower to the mature fruit, while its specialization follows a complex and non-linear behavior during this process. Additionally, by defining sets of genes with different degrees of specialization and applying Gene Ontology enrichment analysis, we identified processes, functions, and components that play a central role in particular fruit development stages. In conclusion, the estimation of diversity, specialization, and specificity summarizes the global properties of the transcriptomes, providing insights that are difficult to achieve by other means.

## 1. Introduction

In 1997, the “transcriptome” was first defined as the set of genes expressed in a given species under particular circumstances [1]. RNA-Seq [2] is the most used method for estimating relative gene expression at the genomic level, i.e., for estimating the transcriptome. For example, in the GEO database [3], there are currently more than 6 million deposited samples (https://www.ncbi.nlm.nih.gov/geo/, accessed on 21 March 2024), many of which are RNA-Seq-based. The main purpose of RNA-Seq experiments is to estimate Differential Gene Expression (DGE) between treatments [4]; however, DGE does not estimate the global changes that occur in the transcriptome.

In [5], we suggested the use of Shannon’s entropy Equation [6] to measure transcriptome diversity, and we derived formulae to estimate transcriptome specialization as well as gene specificities. Analyzing human RNA-Seq libraries as well as microarray results, we showed, for example, that glands have transcriptomes with a low diversity but a high specialization, enhancing the understanding of the global expression profiles in different human tissues. Later, we demonstrated that cancer reduces transcriptome specialization [7], opening the possibility to detect molecular markers for malignant tumors.

The method presented in [5] has been employed, for example, in metabolomics [8,9], lipidomics [10], and evolutionary dynamics [11], as well as to study biochemical diversity [12] and single-cell transcriptomics [13], among others.

RNA-Seq-curated data consist of a matrix of read counts where columns are libraries and rows are genes, loci, or other entities such as gene splicing variants. In this context, the diversity of a library can be estimated by the Shannon’s entropy formula as $H=-\sum p_{i}\log_{2}\left(p_{i}\right)$, where $p_{i}$ is the relative frequency of the *i*th gene (see Equation (1) in [5]). However, as clarified by [14], it is better to express diversity as the base of a logarithm, 2 in this case, raised to the *H* power, i.e., $2^{H}$. This measure gives the Effective Number of Loci, ENL $=2^{H}$, and it is homologous to the term Effective Number of Species used in population studies [15].

The power transformation to obtain ENL implies that this quantity can be interpreted as the number of loci that will result in the estimated *H* value, if all loci where expressed at the same relative frequency. For example, assume that H=12 in a case where we have 20,000 expressed genes. The value of ENL $=2^{12}=4096$ implies that by having only 4096 loci, each one with a relative frequency of expression of 1/4096, the transcriptome will have the same *H* value as the one estimated with the relative frequencies of the 20,000 expressed genes. ENL values can be directly interpreted as a diversity measure of the transcriptome, independently of the fact that two or more RNA-Seq libraries come from different conditions (treatments).

On the other hand, in Equation (3) of [5], we defined Locus Specificity (LS) as the information that its expression provides about the identity of the source library, and taking the weighted average of LS for all expressed loci, we also defined library specialization (Equation (4) in [5]).

In summary, by estimating the diversity as ENL, the Locus Specificity (LS) and the specialization of each library in an RNA-Seq, the researcher obtains a perspective of the results which is hidden if only DGE analyses are performed. Furthermore, for complex RNA-Seq experiments, which could involve various factors at different levels, the results from different libraries can be added to perform an in silico analysis, ignoring, in turn, one or more factors or factor levels, obtaining new perspectives of the results. With this aim, we programmed an R [16] package called “infoRNAseq” [17], which is publicly available at https://doi.org/10.5281/zenodo.10462650 (accessed on 21 March 2024), and we performed the analysis of RNA-Seq data with the methods presented in [5], plus extra data and functions which made the analyses easy to perform and interpret.

Here, we analyze the results of a factorial RNA-Seq experiment whose primary aim was to determine the expression changes in developing fruits of chili pepper. The first factor in that experiment was genotype at 12 levels: six domesticated varieties, “D”, of *Capsicum annuum* L.; four wild ancestors, “W”, of the cultivated chili peppers; and two crosses, “C”, between a D and a W genotype performed in both directions. The second factor was the fruit’s development stage, which was sampled every ten days from anthesis to maturation in each one of the genotypes (see Materials and Methods). From the analysis of this experiment, we previously detected a set of genes relevant for the domestication process [18], unraveled the inheritance of gene expression patterns during fruit development in *Capsicum* [19], and developed a method to analyze time expression profiles [20,21,22]. Nevertheless, this experiment has not been previously analyzed with the tools derived from Shannon’s Information Theory, as implemented in the R package “infoRNAseq”.

Here, we performed a comprehensive analysis of the 179 RNA-Seq libraries in the *Capsicum* experiment from the perspective of the tools presented in [5]. The 179 libraries contain a total of more than three billion reads mapped to the *Capsicum* reference genome, and they include replicated libraries for 12 different accessions (genotypes) in different fruit development stages, plus 4 libraries from seedlings from two accessions (see Materials and Methods).

First, we studied the whole panorama at the individual library level, finding that the diversity of the transcriptomes decreases linearly as a function of the time of fruit development, and also that transcriptome specialization presents a complex pattern with time. In the second step, we grouped all libraries from different fruit development times into accessions, disregarding the time factor. Using this approach, we were able to segregate genes by specificity to types of accessions, finding sets enriched in particular Gene Ontology (GO) [23] categories. The same process was carried out for sets of genes exclusively expressed in domesticated (D) or wild (W) accessions. Finally, we summarized the data by the time of fruit development, in this case disregarding the accession of origin; thus, we confirmed the fact that, in general, the diversity of the transcriptomes decreases linearly as a function of the time of fruit development, and we found groups of genes with high specificity in each time of fruit development, for which we detected enriched GO categories. We believe that the results presented here will enhance the understanding of the fruit development in the genus *Capsicum*, as well as in other related Solanaceae.

Following the principles for reproducible research [24], we include all R calculations performed on the data here, and the resulting R objects are publicly available upon request.

## 2. Results and Discussion

In the *Capsicum* experiment, we used a representative sample of chili pepper genotypes, which included both domesticated (D, 6) and wild (W, 4; *Capsicum annuum* var. *glabriusculum*) ancestors of the *annuum* species, as well as two reciprocal crosses (C) between the D and W accessions. Thus, it is reasonable to assume that the results presented here are representative of the majority of the high diversity of *Capsicum annuum* L. genotypes, which is the species of chili pepper of the largest economic importance in the world [25].

### 2.1. Global Analysis per Library

Using the functionalities of the infoRNAseq R package, we estimated the diversity (ENL) and specialization using each one of the 179 libraries as experimental units in the *Capsicum* experiment. A concern in this analysis was if the different numbers of reads of each library, i.e., the depth of sequencing, appreciably affected the estimates of the parameters. To address such concern, we obtained a random sample of reads from each library, using the minimum number of reads among all the libraries (approximately 10.3 million reads) as a sample size. By comparing the results of the whole dataset with those of the reduced sample, we confirmed that there were no relevant differences in the parameters estimated; thus, in our data, a depth of sequencing of 10.3 M reads per library was enough to give statistically robust and reliable estimates of the informational parameters; see Appendix A for details.

Figure 1 presents the plot of Diversity per Specialization in the 179 libraries, annotated by organ and development time.

In Figure 1, the first notable pattern is that 20 out of the 22 points corresponding to libraries from mature flowers at 0 Days After Anthesis (DAA), represented by yellow stars, are located in the upper right-hand-side quadrant; this is in the zone of high diversity and high specialization with reference to the means of the estimates (dashed grey lines). Only one point, corresponding to a fruit in early development (pale green diamond), is included in this quadrant, and only two points, corresponding to flowers, are slightly below the mean of specialization at the right-hand side of the plot. The segregation of all flower points from the fruits in further stages of development is reasonable because it indicates that the transition from flower to fruit has strong consequences in the corresponding transcriptomes, and the zone where the flower points are located suggests the expression of highly specific flower genes, which increases library specialization. We can also notice in the data for flowers in Figure 1 a trend of decreasing specialization as the diversity (ENL) increases. This indicates that a larger ENL in the flower implies, in general, a lower library specialization.

On the other hand, the upper left-hand-side quadrant of Figure 1, corresponding to the zone with diversity below the mean and specialization above the mean, is populated mostly with libraries from fruits in the maturation stages (Time ⩾40 DAA; orange and red diamonds). At these stages, fruit elongation has stopped, and the position of the points in the quadrant indicates that the fruits with ⩾40 DAA have, in general, transcriptomes with a low ENL, which includes many genes with specialization above the average for that parameter.

Furthermore, in Figure 1, we can observe a general trend of decreasing specialization when diversity (ENL) increases in both flowers (shown as stars) and fruits (shown as diamonds). This indicates that for cases with a small ENL, there is a tendency to have high specialization; for example, the three orange diamonds to the left-hand side of the plot, corresponding to fruits at 40–60 DAA, have the lowest diversity—ENL near 200—while their specialization is near 0.8, and in contrast, the two points for fruits with the largest ENL, the green diamonds at the right-hand side of all fruit symbols and corresponding to 30 and 20 DAA have a large ENL but a low specialization. It is difficult to give a straightforward interpretation to these results because we have three dimensions in the plot: ENL, specialization, and time (DAA). For a direct analysis, we present plots of ENL × Time and specialization × Time in Figure 1 and Figure 2, respectively.

Previously, in [26], using de novo assembling for the transcriptome of a single accession (ST), and sampling only at 10, 20, 40, and 60 DAA, we observed that the specialization increased from 20 to 60 DAA and that diversity presented a global minimum at 60 DAA in that accession. These results are consistent with the general pattern shown in Figure 1 for the full set of accessions.

#### 2.1.1. Transcriptome Diversity Decreases as a Function of Time in *Capsicum* Fruits

To appreciate the change in diversity (ENL) as a function of the time of fruit development, Figure 2 shows a plot of the estimates of ENL in the 175 libraries from flowers and fruits, annotated by type of accession.

In Figure 2, the number of points plotted per time of fruit development is not homogeneous; for times from 0 to 60 DAA, we have 24 points per time (which correspond to replicated libraries at those times in each one of the 12 accessions), while for 70 DAA, there are 6 points, and, finally, for 80 DAA, we have only 2 points. This is due to the fact that maturation took longer than 60 DAA for three D accessions: CW, AS, and JE. All the three accessions reached 70 DAA at maturation, while only AS reached 80 DAA at full maturation (see Materials and Methods).

Furthermore, in Figure 2, we can appreciate how the means of ENL per time of development, shown as circles with a plus sign embedded, decrease following a linear trend. In the mature flowers, at 0 DAA, the mean of ENL is ≈6296, while in the mature fruits of the AS accession at 80 DAA, the mean has its lowest value, ≈2635, a net decrease of 3661 effective loci during the entire fruit development process. We fitted a linear model for ENL as a function of time (black line in Figure 2), and the model has an estimated intercept of 6197 with a slope of −46, indicating that, on average, the ENL steadily decreased by almost 50 loci per day of fruit development. The linear model is highly significant (*p*-value $<2\times 10^{-16}$), and 62% of the ENL variance is explained by the time of development ($R_{adj}^{2}\approx 0.62$), while the residuals of the model follows a normal distribution (*p*-value 0.3375 in the Shapiro–Wilk test; see Appendix A for details).

Figure 2 also allows us to examine the ENL variation at different times of development. The variation is highest at 40 DAA (standard deviation $\hat{S}\approx 1211$), when the fruit stops growing and begins to mature in most accessions, and it is lower within the time points that occur in all 12 accessions at 20 DAA, $\hat{S}\approx 444$, when the fruit is actively growing. Furthermore, we can notice that at 0, 10, 30, 40, 50, and 60 DAA, the largest ENL occurs in D accessions, and the opposite happens only at 20 DAA.

The data and model presented in Figure 2 show that transcriptome diversity in *Capsicum* steadily decreases as the time of fruit development advances. This implies that the flower has a more complex and heterogeneous transcriptome than the developing fruit, suggesting that the intricacy of the genetic interactions decreases as fruit development progresses. This has been indirectly observed during fruit development in other species, such as apple [27], Chinese bayberry [28], raspberry [29], and other crops whose fruit development was analyzed using RNA-Seq [30].

#### 2.1.2. Transcriptome Specialization in *Capsicum* Fruits Varies with Time

Figure 3 shows the plot of library specialization per time in the 175 libraries from flowers and fruits, annotated by type of accession.

In contrast to Figure 2, where the diversity per time shows a linear relation, Figure 3 exhibits a more complex pattern for library specialization with time. Mean specialization per time, shown as a circle with a plus sign embedded, presents a global maximum of 0.7 at the mature flower (0 DAA) and a global minimum of ≈0.39 at 20 DAA, increasing to a local maximum of ≈0.56 at 50 DAA. The 4th-degree polynomial regression model, shown as a black line in Figure 2, is highly significant (*p*-value $<2\times 10^{-16}$). However, it explains less than 50% of the variation of specialization as a function of time ($R_{adj}^{2}=0.4661$). Interestingly, at time points of fruit development where we have data for the 12 accessions, the maxima of specialization occurs in W accessions (blue squares) at 0, 10, 20, and 30 DAA, while the crosses, C (purple diamonds), present maxima for the parameter at 40, 50, and 60 DAA. In no case is there a maxima of specialization in a domesticated accession within the interval of 0 to 60 DAA.

This fact suggests that the domestication process decreases the specialization of the transcriptome during fruit development. The largest dispersion of specialization within the same time occurs at the mature flower ($\hat{S}\approx 0.16$), while the smallest happens at 10 DAA ($\hat{S}\approx 0.04$). Thus, specialization dispersion is 0.16/0.04=4 times more compact in the most active stage of fruit elongation, at 10 DAA, than in the flower, at 0 DAA, and this suggests, again, that the flower has the more complex transcriptome during fruit development (see Appendix A for details).

### 2.2. Analyses by Genotype

To carry out the analysis at the genotype level, ignoring the effect of the time of development, the 175 libraries from flowers and fruits were added by genotype, obtaining a matrix of counts with 12 columns corresponding to the 12 genotypes (accessions). This allowed for the estimation of transcriptome properties taking into account only genetic differences by disregarding the effect of time of fruit development (see Materials and Methods and Appendix A for details of the procedure).

Figure 4 presents the plot of Diversity (as ENL) per Specialization for the 12 genotypes.

Comparing Figure 1, which presents the results at the library level, with Figure 4, which summarizes the results at the genotype level, it is worth noting the changes in the range of the specialization values. In Figure 1, the range of the specialization values goes from a minimum of ≈0.28 up to a maximum of ≈1.04 with a mean of ≈0.50. In contrast, in Figure 4, such range goes from a minimum of ≈0.071 up to a maximum of ≈0.114 with a mean of ≈0.095. This contraction in the specialization range in the analysis at the genotype level is due to the fact that the measure of specialization is the weighted average of loci specificities, and given that specificities for particular times of development are disregarded, transcriptome specialization is drastically reduced. In the analysis at the genotype level, only specificities with reference to particular genotypes were taken into account.

On the other hand, the range of ENL in the analysis at the library level (Figure 1) compared with the one at the genotype level (Figure 4) also changes. In Figure 1, ENL has a minimum of 1942, a mean of 4757, and a maximum of 7423 (a net change of 4757 − 1942 = 2815), while in Figure 4, ENL varies between 4779 and 6401 (a net change of 6401 − 4779 = 1622) with a mean of 5957; this gives a strong displacement to larger values of ENL in the analysis at the genotype level compared with the one at the library level. Again, these changes are due to the fact that the diversity of expression within times at each accession is being ignored.

In Figure 4, we see that there is no significant difference in diversity (ENL) between the accession types (D, W, and C; *p*-value 0.30). This is in contrast to the findings reported in [31], which found a decrease in gene expression diversity in various species of animals and plants, but not in soybean or maize. However, in that publication, the authors measured expression diversity simply as the average of the Coefficient of Variation (CV) of each gene, calculated as the ratio between the standard deviation and the mean of expression values. As mentioned in [32], the use of the CV is meaningful only for variables originally expressed on a ratio or log–interval scale, which is not the case for the expression values. Here, we used ENL, which measure true genome-wide transcriptome diversity and is based in Shannon’s entropy formula, the most profound and useful of all diversity indices [14]. The fact that ENL varies within types of accessions, as shown in Figure 4, means that the expression profiles of each accession are very different, without a clear correlation with accession type.

The differences in specialization (*Y*-axis) between types of accessions shown in Figure 4 are significant (*p*-value 0.009) and mainly due to the fact that the four W accessions are above the specialization mean (horizontal dashed line), while the six D accessions are below the mean. The significant differences in specialization between D (red circles) and W (blue squares) can be explained by the differences in the genetic selection pathways between those accession types. D accessions were selected from their wild ancestors, W, to accommodate human necessities, related to the domestication syndrome [33], while W accessions have evolved to adapt to their particular geographic environments. Thus, it appears reasonable that when measuring specialization in the context of accession types, the D ones are less specialized than the W ones because they share a common genetic selection pathway. In this sense, the closest accessions in the diversity by specialization space in Figure 4 are the ones from the northern Mexican state of Sonora, SR and SY, which show a high diversity and specialization.

Finally, in Figure 4, the $F_{1}$ cross CQ (purple diamond in the upper left-hand side of the plot) is an outlier in both diversity by presenting the lowest ENL and diversity by presenting the highest specialization. That point is far away from the cross between the same parents in opposite direction, $F_{1}$, which has its specialization close to the mean of that parameter and above the average ENL. The large differences in diversity and specialization between the two crosses, CQ and QC, as well as between them and the transcriptomes of their parents, QC and CM, are explained by the surprising heterogeneity in the inheritance of the gene expression patterns, which is discussed in [19].

Locus Specificity (LS; Equation (3) in [5]) varies between 0, when the locus is expressed at the same frequency in all groups studied, and a maximum of $\log_{2}\left(g\right)$, when it is expressed in only one of the *g* groups studied. To analyze LS per accession, we considered which of them presented the maximum expression of the locus and assigned that LS to the corresponding accession. For example, if a given locus presented its maximum expression in accession CW, the LS for that locus will be considered to belong to the group CW, etc. This procedure allowed us to segregate LS per accession, and Figure 5 presents the distributions of relative LS per accession, annotated by type: domesticated (D), cross (C), or wild (W).

In Figure 5, box plots of the relative LS are colored by type (D, C, or W), and, within type, they are ordered by the mean relative LS, indicated by an asterisk. The numbers above the medians give the rounded percentage of loci classified in each distribution. The approximated percentages of loci classified in each accession vary between a minimum of 2 for the cross CQ up to a maximum of 15 for the domesticated accession CW. Interestingly, the largest percentage of loci, 15%, is in the D accession CW, which also has the lowest mean of relative LS, while the opposite happens for the C accession CQ, which has the minimum percentage of loci, 2%, while having the highest mean of relative LS. This is coherent with the results shown in Figure 4, in which CW presents the minimum specialization, while CQ presents the maximum of that parameter.

Furthermore, in Figure 5, we can see that the mean of relative LS in the D type, 0.127 (red horizontal line), is notoriously lower than the means of relative LS in the C and W types, which are 0.189 and 0.188, respectively (dashed horizontal lines). This is also consistent with the results shown in Figure 4, which show larger specialization for W accessions.

LS in the analysis per genotype allowed us to isolate genes which are “generalist” in the sense of being expressed nearly at the same frequency in all genotypes (relative LS ≈0), of those that are highly specific to a given genotype (relative LS ≈1). Here, we defined as generalist genes with relative LS ≤0.05, and with this criterion, approximately 53% of all genes, 17,333/33,007, could be considered as being generalist.

Using Gene Ontology (GO) categories [23], coupled with the R package “Salsa” [20,22], we performed GO enrichment analysis for the set of generalist genes. The GO Cell Component (CC) “transcription factor complex” (GO:0005667) is defined as a protein complex capable of associating with DNA to regulate transcription (https://www.informatics.jax.org/vocab/gene_ontology/GO:0005667, accessed on 21 March 2024). The CC was significantly (*p*-value $\approx 8.5\times 10^{-5}$) enriched in the set of generalist genes, with an estimated odds ratio of ≈3.3, meaning that it was 3.3 times more likely to find genes of the transcription factor complex in this set than the number expected at random. Figure 6 presents the Standardized Expression Profiles (SEPs) for the 51 generalist genes annotated in the GO category GO:0005667, separated by type of accession.

The SEPs plots in Figure 6 are significantly different between D and W genotypes (*p*-value $3.9\times 10^{-15}$), and they show evidence that suggest that the expression of genes in the transcription factor complex are modified by the domestication process. The mean expression of these genes is higher in D compared with W accessions in the flower (at 0 DAA), as well as in the stage of the fastest fruit’s growth, at 10 DAA. This is in line with the fact that the fruit size is much larger in the D genotype than in the W genotype, implying that there is higher and the earliest transcription factor activity in D accessions during the first stages of fruit development. This confirms the previous finding in [18], in which we showed that there is a set of asynchronous genes that have their highest expression level at 10 DAA in D accessions, while the same genes present such maximum later at 30 DAA; most of the genes that code for elements of the transcription factor complex follow this pattern, as can be seen in Figure 6. Furthermore, within the 51 generalist genes which the SEPs show in Figure 6, 14 are Transcription Factors (TF), and of these, 6 (43%) are members of the E2F family (see Appendix A for details). The E2F transcription factors are key regulators of cell cycle progression [34], and the patterns shown in Figure 6 confirm that domestication strongly modifies the dynamics of the process.

To further investigate the differences in gene expression between D and W genotypes, we performed the analysis by grouping fruit libraries by type (see Appendix A for details). By examining LS from this analysis, we were able to find a set of 191 genes exclusively expressed in D accessions, and a set of 157 genes exclusively expressed in W accessions. Figure 7 presents the SEPs for the sets of genes exclusively expressed in each type of accession.

The fact that there is a set of genes exclusively expressed in one type of genotype during fruit development is interesting, because it corroborates the fact that domestication strongly modifies the transcriptome. In Figure 7, we can see that even when the main difference between SEPs of genes exclusively expressed in D or W accessions is larger at the flower (0 DAA), globally, the mean expression in those profiles is highly significant (*p*-value $<5.9\times 10^{-8}$).

Within the genes Exclusively Expressed (EE) in each accession type, we found TF, whose identifiers and descriptions are presented in Table 1.

Of the TF exclusively expressed in the D accessions shown in Table 1, the “Nuclear transcription factor Y subunit B-5” (XP_016567970.1) belongs to the CaNFYB family, which presents promoter *cis*-elements with notable occurrences of light-responsive and stress-responsive binding sites [35]. The “Transcription factor AS1-like” (XP_016542429.1) is significantly expressed only at 0 DAA in the CW accession, which is a sweet chili pepper with large fruits, even when there are also a few reads in libraries from accessions CM, ST, and AS at different times, and orthologous of the AS1 (Asymmetric Leaves 1) in *Arabidopsis* have been reported as important to establish chromatin modifications [36].

Genes of the “Ethylene-responsive transcription factor ERF” family, as XP_016544751.1 (row 4 in Table 1), have been reported to contribute to various processes including plant growth, development, and response to stresses [37]. Interestingly, the ERF transcription factor EPI1 has been shown to be a negative regulator of dark-induced and jasmonate-stimulated senescence in *Arabidopsis* [38], while the family member ERF109—which identifies locus XP_016544751.1 in *Capsicum*—has been reported to mediate cross-talk between jasmonic acid and auxin biosynthesis during lateral root formation [39]. It is interesting that this TF was found expressed exclusively in fruits of the D genotypes, which suggests that it could be playing a role in senescence.

Furthermore, genes like the “scarecrow-like transcription factor PAT1” (XP_016581393.1) participate in phytochrome-B (phyB) signal transduction and could have a role in stress responses [40].

The two TFs exclusively found in W accessions and shown in Table 1, “B3 domain-containing transcription factor ABI3” (XP_016577487.1) and “Zinc finger BED domain-containing protein RICESLEEPER” (XP_016582130.1), have been associated with molecular mechanisms for tolerance to abiotic and biotic stresses [41] and fruit size [42], respectively.

However, genes for TF exclusively expressed in D or W accessions in Table 1 do not present a fully coherent expression pattern in all D or W accessions, which suggests that variations in genotypes are very important in the expression pattern of these genes, and the only common denominator found among them is their role in tolerance to various forms of stress.

### 2.3. Analyses by Time of Development

To perform the analysis at the time of fruit development ignoring the effects of genotype, the 175 libraries from flowers and fruits were added by time of development, obtaining a matrix of counts with nine columns, corresponding to each of the development times. This allows for the understanding of changes in transcriptome properties as a function of the time of fruit development, ignoring the effects of genotypes (see Materials and Methods and Appendix A for details).

Figure 8 presents the trajectory followed by the transcriptomes in the space of Diversity (as “Effective Number of Loci”) × Specialization, for each one of the fruit development times.

In Figure 8, the dashed grey line marks the trajectory followed by the transcriptomes in the space of Diversity × Specialization during fruit development. The path begins in the upper right-hand side of the plot, where the point corresponding to the mature flower (0 DAA) shows the highest specialization and largest diversity. This is the only time point that exists in the quadrant of high diversity and specialization defined by the means of the two parameters (red lines in Figure 8). From the point at 0 DAA and forward in time, the specialization is lower than its mean, except at 50 DAA where the specialization is slightly higher than the mean. Concurrently, from 0 DAA and forward in time, diversity decreases up to its minimum, which is reached at 80 DAA. This is consistent with the plot of Time × Diversity presented in Figure 1.

The positions of the time points in Figure 8 give a glimpse of the global transcriptome properties, while the lengths of the lines connecting neighboring times (grey dashed line segments) inform us about the relative importance of the changes. For example, the change from 0 to 10 DAA is the longest segment of the path, implying the strongest transcriptome change, which is explained by the flower-to-fruit transition. In contrast, the lengths of the segments from 40 to 50 and from 50 to 60 DAA are very small, showing that between 40 and up to 60 DAA, the transcriptomes present an almost steady state with reference to diversity and specialization. Finally, it is important to remember that only thre accessions (“CW”, “AS”, and “JE”) represent the time point at 70 DAA, and only the “AS” accession reached the 80 DAA. Thus, time points at 70 and 80 DAA are only relevant for domesticated accessions with long maturation times.

To analyze the LS per time of development, we considered the times at which each locus presented the maximum expression, and we assigned that LS to the corresponding time. This procedure allowed us to separate LS per time, and Figure 9 presents the distributions of the relative LS per time of fruit development, including the numbers of loci with a large relative LS >0.9, as well as the percentages of the total number of cases for each time.

In Figure 9, we can first notice that the largest distributions are the ones for times 0, 10, and 20 DAA, which include 23, 20, and 11% of all loci, respectively. All further times make up 10% or less of the cases. Furthermore, all nine distributions are skewed to the right, i.e., they have a mean that is larger than the corresponding median, and at all the nine times, there are loci that reach the maximum relative LS; the number of those loci is shown in blue figures at the top of the box plots. As could be expected from the plot in Figure 8, the distribution of the relative LS with the largest mean and median corresponds to 0 DAA, i.e., to flower, corroborating that the time of the largest transcriptome specialization occurs at 0 DAA.

To complete the analysis of the LS per time of fruit development, we performed GO enrichment analysis of genes with high relative LS (>0.9) in the flower (at 0 DAA), as well as in the growing stages, from 10 up to 40 DAA and, finally, at maturity, from 50 to 80 DAA. The results of those GO enrichment analysis are presented in detail in Appendix A, enlightening which Biological Processes (BPs), Molecular Functions (MFs), and Cell Components (CC) are preponderant at each stage of the fruit development.

## 3. Materials and Methods

### 3.1. Plant Materials and RNA-Seq Processing

The plant material consisted in 6 domesticated (D) varieties of *Capsicum annuum* L. obtained from commercial sources, 4 wild (W) accessions of *Capsicum annuum* var. *glabriusculum* from different geographic origins, which are ancestors of the domesticated *C. annuum* varieties [25], plus two crosses (C) in both directions of a D × W genotype. All plants were grown under optimal conditions as described in [18], and to obtain RNA-Seq libraries from fruits in development, whole flowers (at time 0 DAA) or whole fruits (at times >0 DAA) were collected and processed as described in [18].

Table 2 presents the accessions (genotypes) utilized in the *Capsicum* experiment analyzed here, as well as the number of RNA-Seq libraries obtained from each accession and the total number of clean reads mapped to the reference *Capsicum annuum* L. genome.

In Table 2, the first column contains the common name of the accession, followed (in parentheses) by the two character keys used to denote the accession throughout the text and the type of accession: W for wild, D for domesticated, and C for cross. We had 4 W, 6 D, and 2 C, and for each one of these, fruits were collected to obtain biologically replicated RNA-Seq libraries at times of development 0, 10, 20, 30, 40, and 60 DAA. Additionally, D accessions CW, AS, and JE presented maturation times of more than 60 days; thus, these three accessions were also sampled at 70 DAA, and AS was also sampled at 80 DAA. Furthermore, replicated RNA-Seq libraries were obtained from seedlings of the W accession QU and the D accession ST. In summary, we performed a factorial experiment with factors “accession” (12 levels) and “time of fruit development” (6 levels for all accessions and extra levels for the late-maturing accessions), plus data from seedlings for two accessions. These data comprised a total of 179 RNA-Seq libraries, with 175 of them from fruit in development and 4 from seedlings; see column “N Lib.” in Table 2 for the total number of libraries per accession. Raw and processed data from the 179 libraries were deposited in the Gene Expression Omnibus (GEO) [3]; see Data Availability Statement. Table 2 also presents the total number of clean reads mapped to the reference genome for each accession (column “M Reads”).

RNA-Seq library construction and processing are presented in [18].

### 3.2. Data Analyses

Previously, in [20], we implemented a method to analyze time expression profiles as Standardized Expression Profiles or “SEPs” and demonstrated the approach with data from the *Capsicum* experiment (see also [21]). The data and functions to apply these algorithms are in the R package “Salsa” [22] and are employed here to summarize the results of the analyses.

To perform the analysis using the methods in [5], we programmed the R [16] package “infoRNAseq” [17], which includes the data analyzed here, but can also be used with data from any RNA-Seq experiment. The following list summarizes the analysis pipeline, including the section of the Appendix A where details of each step can be consulted.

Sample size sensitivity analysis (Appendix A);Global analysis per library (Appendix A);Analyses by genotype (Appendix A);Analyses by time of development (Appendix A).

Briefly, in (1), we performed an analysis to examine how the sample size, i.e., the number of reads per library, affected the estimated parameters. In this step, the number of reads per library was artificially reduced to the minimum found in all the 179 libraries, which is ≈10.3 million reads, confirming that the estimated parameters were not substantially altered by the reduction in the sample size (see Appendix A).

In (2), we performed the analysis using, as experimental unit, each one of the 179 RNA-Seq libraries (see Appendix A). To carry out the analysis per genotype in step (3), we added, by locus (row), the counts from each one of the libraries containing data from fruits in development of each one of the 12 genotypes. Thus, in this step, the experimental unit is each one of the accessions, and the time of fruit development is disregarded (see Appendix A). Conversely, to ignore the effects of genotype, in step (4), we added, by locus (row), the counts from each one of the different times of fruit development, obtaining groups influenced only by the time factor over all genotypes (see Appendix A).

Additionally, in steps (2), (3), and (4), we performed GO enrichment analyses of genes presenting extreme values of relative specialization in either sets showing a generalist tendency by having very low specialization or, on the contrary, by having very high specialization in specific groups of accessions—in (3), or at stages of fruit development—in (4). In all cases, we converted the original *p*-values of the tests into *q*-values by the method in [43] and fixed a minimum threshold on the *q*-values to obtain a reasonable False Discovery Rate (FDR). Tables with the results of the GO-enriched terms are presented and briefly discussed in the corresponding sections of the Appendix A.

## 4. Conclusions

Estimating RNA-Seq libraries’ diversity and specialization summarizes the global properties of the transcriptomes in a two-dimensional space, which allows for understanding, at a glance, the changes that occur in tens of thousands of genes as a result of different factors—in the case of the *Capsicum* experiment, different genotypes and times of fruit development. On the other hand, the estimation of locus specificities defines sets of genes which are generalists—by being expressed at approximately the same frequencies in sets of treatments—or, conversely, specialized genes that are typical of very specific conditions.

Furthermore, by grouping sets of RNA-Seq libraries, we demonstrated how to isolate single factors from complex factorial experiments. We argue that the facilities in the “infoRNAseq” R package represent a methodological advancement which enables the discovery of otherwise hidden aspects of the transcriptome.

From the broad sampling of chili pepper fruits of different genotypes, we conclude that the expected diversity of the transcriptome decreases as a linear function of time from the flower up to the mature fruit. This implies that the expected number of ENL—estimated by the means per time of fruit development—steadily decreases during fruit development and indirectly suggests that the complexity of the transcriptomes is reduced as the time of fruit development advances. We also conclude that the specialization of the transcriptomes exhibits a complex and non-linear behavior during fruit development in *Capsicum*.

Furthermore, from GO enrichment analyses, we were able to detect Biological Processes (BPs), Molecular Functions (MFs), and Cell Components (CC) which play a central role in particular genotypes or fruit development stages.

Finally, it is important to mention that this study was focused only on the differences in gene expression (transcriptional regulation) during fruit growth, development, and ripening, but some other mechanisms are surely involved, like those related to epigenetic modifications (alterations of chromatin structure and the consequent stable changes in gene expression without modifications of DNA sequences) caused by the influence of environmental factors such as light, temperature, and stress conditions, but also due to changes in phytohormones which lead to dramatic changes in color, texture, flavor, and aroma [44,45,46,47,48,49,50,51]. Changes in the epigenome during the growth, development, and ripening of chili pepper fruits have been reported [52], and the main findings have been that DNA hypomethylation was demonstrated to occur in the upstream region of the transcription start site of some methyltransferases (*MET*) and chromomethylases (*CMT*) genes involved in ripening at the breaking stage, causing up-regulation (*CaMET2-like*) or down-regulation (*CaMET1-like1*, *CaMET1-like2*, *CaCMT2-like*, and *CaCMT4-like*). Moreover, DNA hypomethylation repressed the expression of auxin and gibberellin biosynthesis-related and cytokinin-degradation-involved genes, but it induced the expression of ABA biosynthesis-related genes. It will be interesting to extend epigenetic studies to some other aspects of chili pepper fruit growth, development, and ripening in future projects.

## Figures and Tables

**Figure 1 plants-13-00983-f001:**
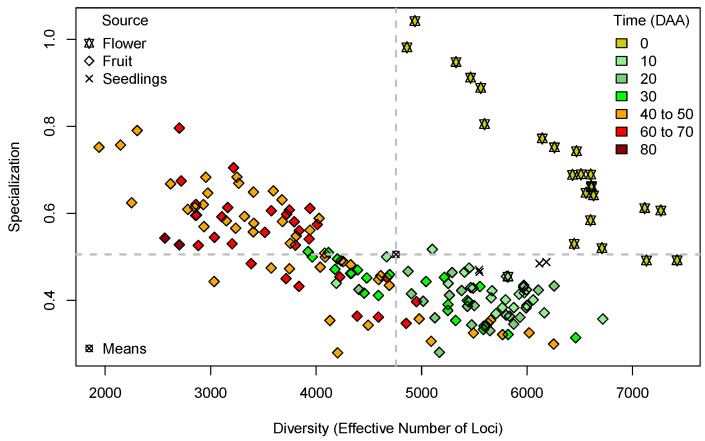
Plot of Diversity (as “Effective Number of Loci”, ENL, *X*-axis) per Specialization (*Y*-axis) for the 179 libraries in the *Capsicum* data annotated by organ and development time. Quadrants (dashed grey lines) are defined by the means of the estimates of Diversity and Specialization.

**Figure 2 plants-13-00983-f002:**
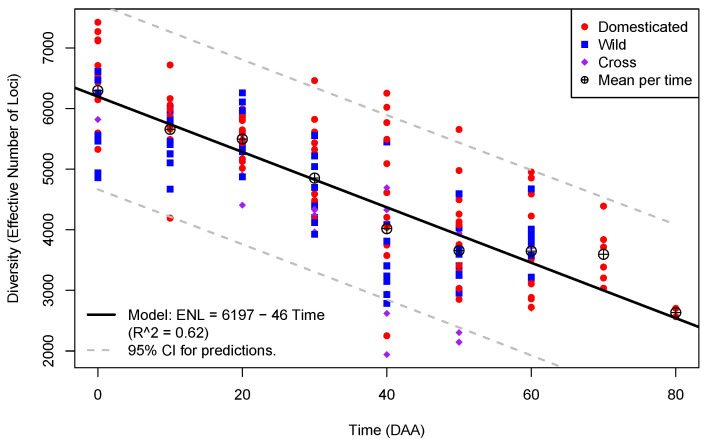
Plot of Diversity (as “Effective Number of Loci”; *Y*-axis) per Time in DAA (*X*-axis) for the 175 libraries from fruits in the *Capsicum* data annotated by type of accession.

**Figure 3 plants-13-00983-f003:**
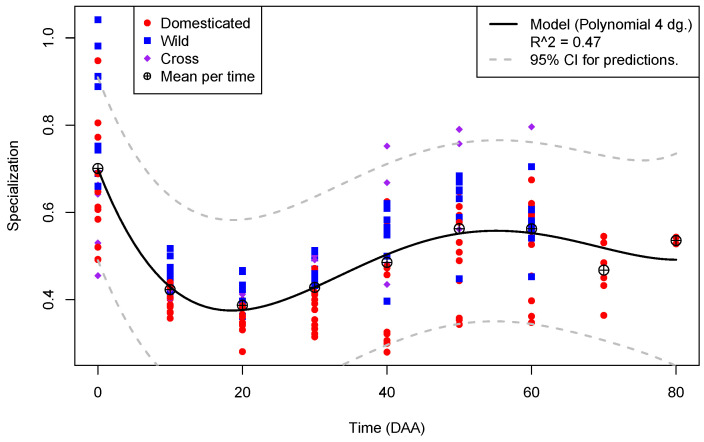
Plot of Specialization (*Y*-axis) per Time in DAA (*X*-axis) for the 175 libraries from fruit in the *Capsicum* data annotated by type of accession.

**Figure 4 plants-13-00983-f004:**
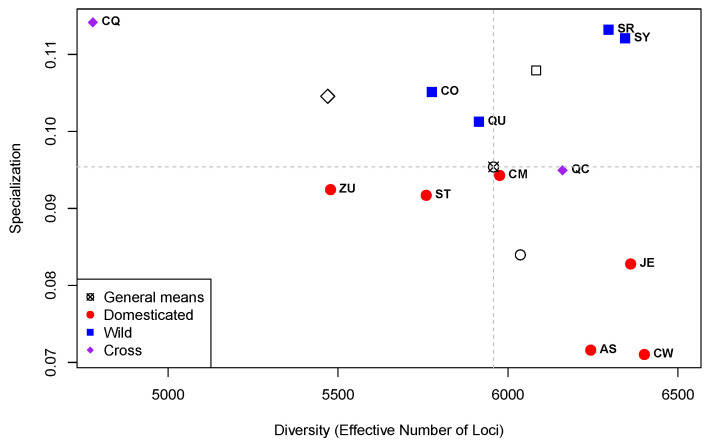
Plot of Diversity (as “Effective Number of Loci”; *X*-axis) per Specialization (*Y*-axis) for the 12 groups of libraries per genotype in the *Capsicum* data. Quadrants (dashed grey lines) defined by the general means of the estimates of Diversity and Specialization and non-colored symbols are the means per type of accession.

**Figure 5 plants-13-00983-f005:**
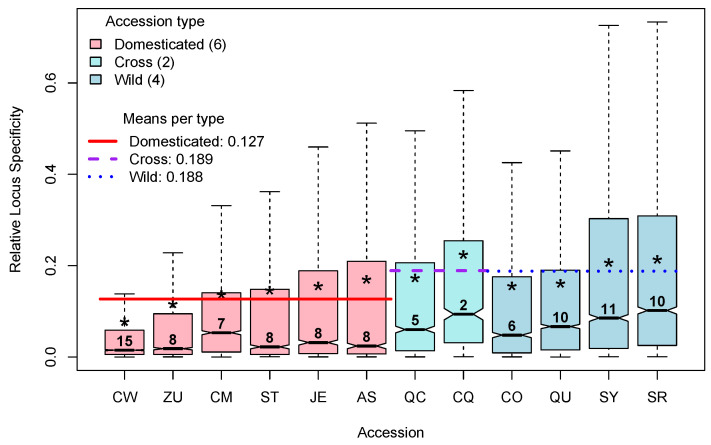
Distributions (as box plots) of the Relative Locus Specificity per accession. Asterisks point to the means of each accession. Numbers above each median are percentages of each class (% of the total of 33,007 loci). Colored lines show the means of each group.

**Figure 6 plants-13-00983-f006:**
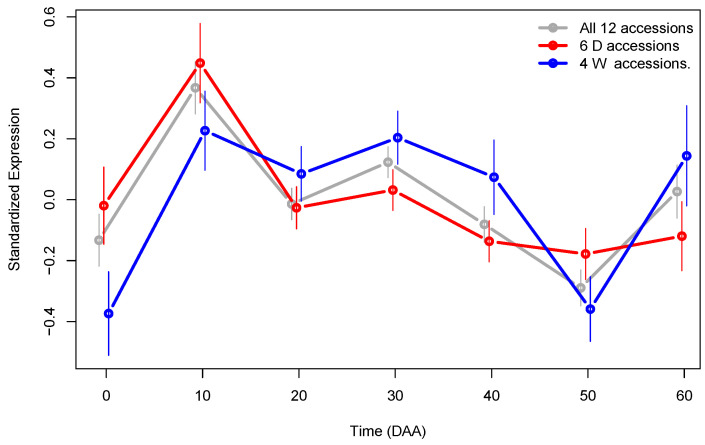
SEPs plot for the 51 generalist genes enriched in the Cell Component “transcription factor complex” (GO:0005667) group by accession type. The 95% CI for mean expression at each time is presented as vertical lines.

**Figure 7 plants-13-00983-f007:**
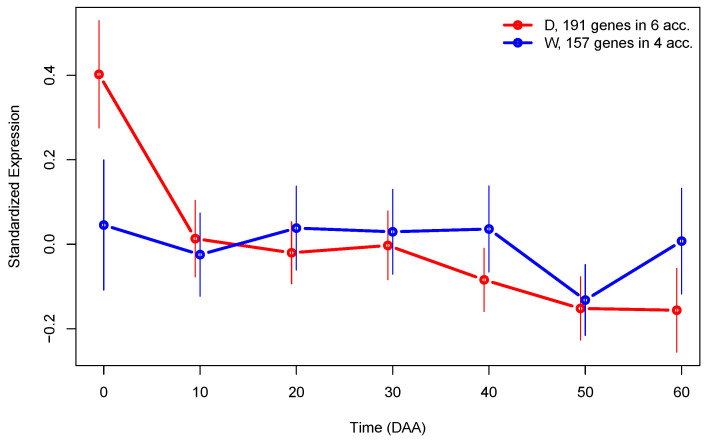
SEPs plots of genes exclusively expressed in domesticated (D, red line) or wild (W, blue line) accessions. The 95% CI for mean expression is presented as vertical lines.

**Figure 8 plants-13-00983-f008:**
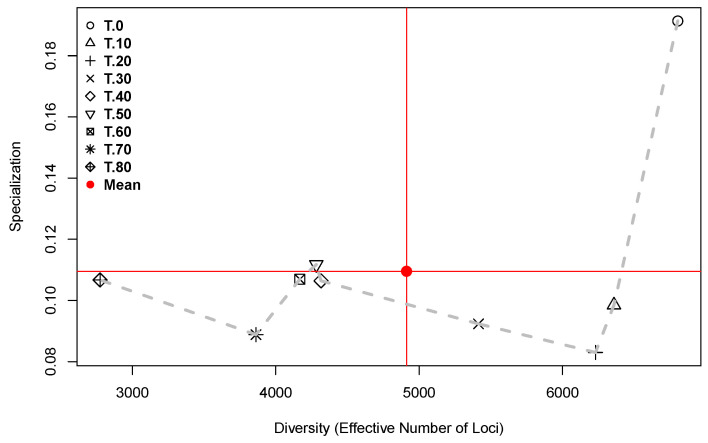
Diversity as “Effective Number of Loci” (*X*-axis) × Specialization (*Y*-axis) for each of the fruit development times for all accessions. Time points in DAA are annotated in the legend. The dashed grey line connects neighboring times.

**Figure 9 plants-13-00983-f009:**
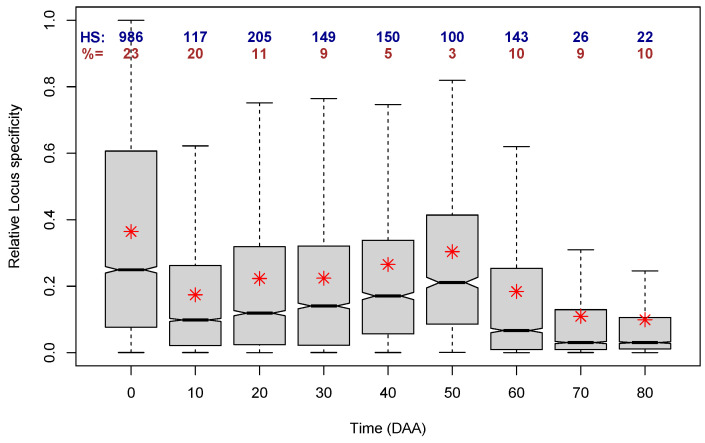
Distributions (as box plots) of the Relative Locus Specificity per time of fruit development over all accessions. Red asterisks indicate the mean of each distribution. Blue numbers (HS:) indicate the number of loci with Relative Specificity > 0.9. Brown numbers (%=) indicate the percentage of the total number of cases per time of development (% of the total of 33,007 loci).

**Table 1 plants-13-00983-t001:** Transcription Factors (TF) found within genes Exclusively Expressed (EE) in D or W.

EE in	Identifier	Description
D	XP_016540811.1	Transcription factor TGA6
D	XP_016567970.1	Nuclear transcription factor Y subunit B-5
D	XP_016542429.1	Transcription factor AS1
D	XP_016544751.1	Ethylene-responsive transcription factor ERF109
D	XP_016581393.1	Scarecrow-like transcription factor PAT1
W	XP_016577487.1	B3 domain-containing transcription factor ABI3
W	XP_016582130.1	Zinc finger BED domain-containing protein RICESLEEPER 2

**Table 2 plants-13-00983-t002:** Accessions (genotypes), number of RNA-Seq libraries, and totals of reads.

Name (Key; Type)	N Lib. ^1^	M Reads ^2^
Piquín Coahuila (CO; W)	14	232
Piquín Queretaro (QU; W)	16	248
Piquín Sonora Red (SR; W)	14	224
Piquín Sonora Yellow (SY; W)	14	221
Ancho San Luis (AS; D)	18	296
Criollo de Morelos (CM; D)	14	250
California Wonder (CW; D)	16	294
Jalapeno Espinalteco (JE; D)	16	274
Serrano Tampiqueño (ST; D)	16	233
Zunla-1 (ZU; D)	14	226
$F_{1}$: CM × QU (CQ; C)	14	279
$F_{1}$: QU × CM (QC; C)	13	236
Sum:	179	3013

^1^ Total number of RNA-Seq libraries. ^2^ Millions of clean reads mapped to the *Capsicum* genome in the libraries.

## Data Availability

The original RNA-Seq data have been deposited in NCBI’s Gene Expression Omnibus [3] and are accessible through the GEO Series accession number GSE165448: https://www.ncbi.nlm.nih.gov/geo/query/acc.cgi?acc=GSE165448 (accessed on 21 March 2024). All curated data and functions for the analyses are publicly available in the R packages “infoRNAseq” [17]: https://zenodo.org/records/10462650 (accessed on 21 March 2024) and “Salsa” [22]: https://zenodo.org/record/7602359 (accessed on 21 March 2024).

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
