# Peer review of "Estimating Transcriptome Diversity and Specialization in Capsicum annuum L."

_plants, 2024, doi:10.3390/plants13070983_

Round 1

Reviewer 1 Report

Comments and Suggestions for Authors

Dear Authors,

I appreciate your work, and here are some comments.

Have you submitted your transcriptome read as SRA in NCBI? If not, why?

Please alter the title as you like: Estimating Transcriptome Diversity and tracing the fluctuation of expression pattern from flowering to maturity in Capsicum annuum L.

Have you provided the link and codes to access the package through Github etc?

Are these genotypes natural populations or varieties? Please mention it clearly. 

Whether the transcriptomes are assumed to be from a specific stage during development or from specific tissues? Does the epigenetic factor not influence the transcriptomes or genes of transcriptomes?

Is it concerned with polymorphic loci and the allele frequency of specific genes among the genotypes studied or populations studied? 

Even if you disregard the time of developmental stages, the complex pattern of transcriptomes can anyhow be mentioned as transcriptome specialization during development, isn't it? Accession differences would provide genotype-specific expressions or specific allele expressions. But how can you define the variations of transcriptome differences during development from fruit to flowering, irrespective of a specific genotype, when you compare global transcription across genotypes? 

Is there a specific scientific name of the wild variety?

At line 130, what do you mean by decreasing specialization? You mean to say the number of transcriptome specialization is decreasing while the diversity of gene expressions across the transcriptomes is increasing?

Please write differently: during the maturity of fruit genes (or alleles of the genes), the specific transcriptome specilization is expected to be lower, while the diversity of these genes or alleles is higher in terms of expression from ENL.

At Figure 2, Why is there no ENL detected after 60 DAA in wild compared to domesticated? Are there fruit sizes and maturity days that correspond to the level of transcriptome expressions from domesticated to wild?

Perhaps you would have selected some specific genes in flowering and fruiting besides the transcriptome analysis? For instance florigen encoding  Flowering Locus T  gene etc.

At Figure 5 Boxplot, How have you calculated percentage of loci for each genotype from total loci, while seems the mean is higher in CQ with lower lower RLS (2%)?

Only you have found Transcription Factors (TF), not any other target downstream genes or genes targeted by miRNA or mRNA expression itself?

Please write precisely. 

Comments on the Quality of English Language

Reviewer 2 Report

Comments and Suggestions for Authors

The genus Capsicum is one of the most important crops and to study its genetic diversity is really necessary. The authors of the articles present an extensive study that brings new interesting information. Unfortunately, the large amount of information makes the article disorganized and confusing. To understand the connections and context, it is required to repeatedly change checking the info in the Manuscript and in the Supplementary.

If the article is supposed to be a comprehensive study, it needs to interconnect the details it brings. I recommend limiting the scope of the work's objectives and making them specific to the selected area. The extent of the Supplementary should be reduced. Alternatively, divide the article according to sub-topics.

The authors do not follow the format required by the editors.

Specific comments:

• Edit the citation of the website (Lines 23; 63; 301)

3. Materials and Methods

3.1. Plant Materials and RNA-Seq Processing

• The origin and sources of plant material are not defined, please define

4. Analyzes by time of development: it is not clear how the plant samples were taken and in what conditions they grew.

English: needs minor revisions, ther are some misspellings, missing words. The language is informal in some places and might be replaced by a more formal way of expressing.

Line 27 – In a research/study  [5] …

Line 28 – formulae – misspelled

Round 2

Reviewer 1 Report

Comments and Suggestions for Authors

Dear Authors,

I appreciate your revisions. However, one point has to be considered: during the development stage towards fruiting, it is expected that mRNA expression and diversity naturally decrease due to senescence during maturation. This is fundamental, but validating the hypothesis with the study providing the understanding is evidential. Further, Exclusively Expressed Gene Analysis showed that TFs like ERF were reported earlier for senescence, and your study is in accordance, for instance: https://www.ncbi.nlm.nih.gov/pmc/articles/PMC6637254/

But inclusion of epigenetic factors would provide further understanding of your further research to see GXE interactions during development in terms of time from DAA to maturity.  Your revisions and responses are acceptable. 

If you wish, make it slightly more precise, and you can discuss slightly concerning the role and importance of epigenetic factors in mRNA expressions besides the development and time.

Reviewer 2 Report

Comments and Suggestions for Authors

Comments have been addressed.

The attractiveness of the article remains the same.

Persistent comments:

Edit the citation of the website (Lines 23; 63; 301). It is not common to mention http.www........ in the text.

The text lacks a link to "SI.5. Appendix: Boxes with R code"

Round 3

Reviewer 2 Report

Comments and Suggestions for Authors

Comments have been addressed.

The attractiveness of the article remains the same.